# The Janus Kinase Inhibitor Ruxolitinib Prevents Terminal Shock in a Mouse Model of Arenavirus Hemorrhagic Fever

**DOI:** 10.3390/microorganisms9030564

**Published:** 2021-03-09

**Authors:** Mehmet Sahin, Melissa M. Remy, Doron Merkler, Daniel D. Pinschewer

**Affiliations:** 1Department of Biomedicine—Haus Petersplatz, Division of Experimental Virology, University of Basel, 4009 Basel, Switzerland; mehmet.sahin@unibas.ch (M.S.); meliremy@gmail.com (M.M.R.); 2Department of Pathology and Immunology, Division of Clinical Pathology, University Hospital of Geneva, 1211 Geneva, Switzerland; doron.merkler@unige.ch

**Keywords:** Arenavirus hemorrhagic fever, LCMV, Janus kinase, JAK inhibitor, Ruxolitinib, nitric oxide, iNOS, microvascular leak

## Abstract

Arenaviruses such as Lassa virus cause arenavirus hemorrhagic fever (AVHF), but protective vaccines and effective antiviral therapy remain unmet medical needs. Our prior work has revealed that inducible nitric oxide synthase (iNOS) induction by IFN-γ represents a key pathway to microvascular leak and terminal shock in AVHF. Here we hypothesized that Ruxolitinib, an FDA-approved JAK inhibitor known to prevent IFN-γ signaling, could be repurposed for host-directed therapy in AVHF. We tested the efficacy of Ruxolitinib in MHC-humanized (HHD) mice, which develop Lassa fever-like disease upon infection with the monkey-pathogenic lymphocytic choriomeningitis virus strain WE. Anti-TNF antibody therapy was tested as an alternative strategy owing to its expected effect on macrophage activation. Ruxolitinib but not anti-TNF antibody prevented hypothermia and terminal disease as well as pleural effusions and skin edema, which served as readouts of microvascular leak. As expected, neither treatment influenced viral loads. Intriguingly, however, and despite its potent disease-modifying activity, Ruxolitinib did not measurably interfere with iNOS expression or systemic NO metabolite levels. These findings suggest that the FDA-approved JAK-inhibitor Ruxolitinib has potential in the treatment of AVHF. Moreover, our observations indicate that besides IFN-γ-induced iNOS additional druggable pathways contribute essentially to AVHF and are amenable to host-directed therapy.

## 1. Introduction

Several members of the arenavirus family, most prominently Lassa virus (LASV), the causative agent of Lassa fever but also several South American arenaviruses such as Junin virus in Argentina, Machupo virus in Bolivia and Guanarito virus in Brazil, can cause viral hemorrhagic fever in humans [1]. In terminal disease stages, manifestations of arenaviral hemorrhagic fever (AVHF) commonly include hypotension and hypovolemia, which culminate in multi-organ failure and a terminal, uncontrollable shock syndrome due to microvascular leak [2,3]. In contrast to other viral hemorrhagic fevers, however, bleeding is less common in AVHFs and in Lassa fever is considered rare. A so-called “cytokine storm” is evident in excessive systemic levels of inflammatory mediators such as nitric oxide (NO), interferon-gamma (IFN-g), tumor necrosis factor (TNF), interleukin-6 (IL-6), IL-8, IL-10, IL-12 and of chemotactic agents such as CXCL10 and CCL2 [1,4]. However, the individual contribution of each one of these factors to microvascular leak and mortality in AVHFs remains to be further investigated.

We have previously reported on AVHF pathogenesis studies in mice expressing human/mouse chimeric HLA-A2 instead of mouse MHC class I molecules (HHD mice), which renders them susceptible to Lassa fever-like systemic disease upon infection with either LASV or the WE strain of lymphocytic choriomeningitis virus (LCMV) [5,6]. The latter virus is genetically closely related to LASV, is not normally pathogenic for healthy adults but causes AVHF in both non-human primates and HHD mice and therefore has been used as a model organism to investigate mechanisms of AVHF pathogenesis without the need for BSL-4 laboratory containment [6,7]. The investigations in HHD mice revealed that the terminal shock syndrome was T cell-dependent and IFN-γ driven and correlated with macrophages activation. Genetic deletion of inducible NO synthase (iNOS), the NO synthase of activated macrophages, in HHD mice prevented disease, identifying NO as a key mediator of the shock syndrome in AVHF. Similarly, T cell depletion prior to LCMV infection or IFN-γ blockade during infection repressed hepatic iNOS and prevented disease despite an otherwise fully-fledged cytokine storm and uncontrolled high-level viremia. These findings identified new promising host-directed strategies for the treatment of AVHF in humans. IFN-γ blocking antibody therapy has been FDA-approved for the treatment of hemophagocytic lymphohistocytosis, a potentially lethal immune hyperactivation syndrome of mixed etiology, and has shown preliminary yet promising efficacy signals in Crohn’s disease [8,9]. IFN-γ receptor downstream signaling involves the Janus kinase (Jak) as well as the signal transducer and activator of transcription (STAT) [10], and a JAK1/2 inhibitor, Ruxolitinib, has been approved for the treatment of polycythemia vera, myelofibrosis and acute graft-versus-host-disease (GVHD) [11]. Interestingly, preclinical models as well as preliminary clinical evidence suggest Ruxolitinib may be effective in hemophagocytic lymphohistocytosis [12], and recent case series suggest the drug may also reduce disease severity in COVID-19 patients [13]. A second independent yet promising path to host-directed therapy in AVHF consisted of the blockade of TNF, which besides IFN-γ signaling represents a key pathway of macrophage activation [14]. Accordingly, TNF blockade has been shown to reduce mortality in murine models of Dengue virus pathogenesis [15,16].

Here, we investigated the efficacy of Ruxolitinib and of anti-TNF antibody in preventing terminal disease in the AVHF model of LCMV-infected HHD mice. Our findings position Ruxolitinib but not TNF blockade as a promising strategy for the prevention of microvascular leak in AVHF treatment, and suggest druggable pathways other than those leading to iNOS activation also have an essential contribution to pathogenesis.

## 2. Materials and Methods

### 2.1. Mice and Ethics Statement

HHD mice carrying a human-mouse chimeric HLA-A2.1 transgene, in conjunction with homozygous deletions of the beta-2-microglobulin and H-2Db genes [17], were used for all the animal experiments. The mice were on a C57BL/6 background. Colonies were maintained at the Laboratory Animal Science Center of the University of Zurich, Switzerland. All animal experiments were conducted in the University of Basel in accordance with the Swiss law for animal protection and with authorization by the Cantonal Veterinary Office of the Canton of Basel. Breedings and experiments were performed under specific pathogen-free conditions. Animals were assigned to experimental groups based on sex- and age-matching. Mice undergoing infection experiments were monitored daily from day 6 to day 10 after infection for evidence of terminal disease. Ruffled fur, lethargy and hunchback served as humane endpoints.

### 2.2. Virus Titration and Infection

The LCMV strain WE originates from F. Lehmann-Grube (Heinrich-Pette Institute, Hamburg, Germany) and was obtained from R. Zinkernagel and Hans Hengartner (University of Zurich, Switzerland). The virus was administered intravenously (i.v.) at a dose of 200 PFU.

Virus stocks were grown on L929 cells (obtained from the American Type Culture Collection). Virus stocks as well as virus loads in infected mouse blood samples were titrated by immunofocus assay as previously described [18,19]. In brief, virus-containing samples were plated at ten-fold serial dilution and 3T3 cells were added. After 2 h of incubation at 37 °C, DMEM containing 1% methylcellulose was added as overlay. Two days later, the supernatant was discarded, and the cells were fixed with 4% paraformaldehyde for 30 min. Then, fixed cells were permeabilized with phosphate-buffered saline (PBS) containing 1% TritonX100. After blocking with 5% fetal calf serum, primary antibody (VL4 rat anti-LCMV-NP [18], was added, followed by the detection antibody (horseradish peroxidase-conjugated goat-anti-rat-IgG, Jackson Immunoresearch, West Grove, PA, USA). DAB reagent (Dako, Santa Clara, CA, USA) was used for the color reaction. Infectious titers are reported as focus forming units (PFU). Virus work was performed under biosafety level 2 laboratory conditions in accordance with Swiss guidelines.

### 2.3. Treatment with Ruxolitinib, Anti-TNF and Anti-IFN-γ

Ruxolitinib (INC18424, Selleckchem, Houston, TX, USA) was dissolved in either 0.5% methylcellulose or PBS supplemented with 0.1% Tween-20. It was administered at a dose of 60 mg/kg by gavage every 12 h from day 3.5 to 8 post-infection. Vehicle controls were given PBS supplemented with 0.1% Tween-20. For in vivo blockade of TNF and IFN-γ, respectively, anti-TNF (XT3.11, BioXcell, Lebanon, NH, USA) and anti-IFN-γ neutralizing antibodies (XMG1.2, BioXcell, Lebanon, NH, USA) were administered intraperitoneally at doses of 1 mg and 2 mg per animal, respectively, on day 5 post infection.

### 2.4. Determination of Body Temperature, Pleural Effusions and Skin Edema

The body temperature of mice was measured using a thermometer with a mouse rectal probe (Bioseb, Vitrolles, France). To determine pleural effusion volumes, pleural fluid was collected from the thoracic cage using a syringe with a 23-gauge needle. To quantify skin edema the wet/dry ratio was determined. In brief, ventral skin flaps of approximately 4 cm^2^ were dissected after euthanasia of the animals and were weighed, then were incubated at 60 °C for 48 h and weighed again to calculate the ratio of wet/dry weight.

### 2.5. NOx Determination

The nitric oxide metabolites nitrite (NO_2_^−^) and nitrate (NO_3_^−^) were measured in mouse serum using the Nitrate/Nitrite Colorimetric Assay Kit (Cayman, 780001, Ann Arbor, MI, USA). The colorimetric reaction was quantified at 540 nm absorbance in a plate reader (Sapphire Bioscience, Waterloo, NSW, Australia).

### 2.6. iNOS mRNA Expression Analysis

Total cellular RNA was extracted from organs using TRI Reagent (Sigma Aldrich, St. Louis, MO, USA). iNOS mRNA and the housekeeping gene GAPDH mRNA were quantified in parallel for normalization. The following 20X kits from Applied Biosystems (Foster City, CA, USA) were used for iNOS and GAPDH respectively: Mm00440502_m1 (Nos2, FAM) and Mm00484668_m1 (Gapdhs, VIC). RT-PCR was performed with Superscript III One Step Platinum Taq (Invitrogen, Carlsbad, CA, USA) using the manufacturer’s recommendation for assay preparation and cycling.

### 2.7. Histology and Immunohistochemistry

Mouse tissues were fixed in 4% paraformaldehyde and embedded in paraffin. Sections were then processed for immunohistochemistry as follows: endogenous peroxidases were inactivated by PBS supplemented with 3% hydrogen peroxide and slides were incubated with PBS containing 10% fetal calf serum to reduce non-specific binding. The sections were then incubated with the following primary antibodies: rat anti-human CD3 (clone CD3-12, known to cross-react with murine CD3 on mouse T cells, AbD Serotec, Kidlington, UK) or anti-iNOS (Assay Designs, Ann Arbor, MI, USA) or anti-LCMV-NP (VL-4, [18]). Then, sections were stained with biotinylated secondary antibodies specific for rat (DakoCytomation, Glostrup, Denmark). For detection, ExtraAvidin-Peroxidase (Sigma Aldrich, St. Louis, MO, USA) was used. Bound secondary antibody was revealed with 3,3′-diaminobenzidine as chromogen (DakoCytomation, Glostrup, Denmark) to visualize with brightfield microscopy. Slides were scanned using a MIRAX Midi slide scanner (ZEISS, Oberkochen, Germany) at 200× magnification.

### 2.8. Statistical Analysis

The GraphPad Prism software (version 8, GraphPad Software, San Diego, CA, USA) was used for all analyses. Statistical significance in single measurement comparisons of more than two groups were assessed by one-way ANOVA followed by Bonferroni’s post-test for multiple comparisons. When multiple measurements of two or more groups were compared, a two-way ANOVA and Bonferroni’s post-test for multiple comparisons were performed. For the statistical evaluation of differences in disease incidence rates Fisher’s exact test of the GraphPad QuickCalcs online tool was used. Statistical significance of each comparison was evaluated by calculating a two-tailed *p* value. The *p*-values <0.05 were considered statistically significant (indicated as * in figures). Differences with *p* > 0.05 were considered not statistically significant (“ns” or no indications in figures).

## 3. Results

### 3.1. JAK Inhibitor Ruxolitinib but Not TNF Blockade Prevents Terminal Disease in LCMV-Infected HHD Mice

To assess the efficacy of Ruxolitinib and TNF-blocking antibody therapy, respectively, in preventing AVHF, we treated LCMV-infected HHD mice with Ruxolitinib from day 3.5 after infection until the peak of disease in control animals (day 8) or gave TNF-blocking antibody on day 5 (Figure 1A). The latter time point was chosen for antibody administration since IFN-γ blockade on day 5 consistently prevented disease in our model (Table 1, [6]). LCMV-infected control groups were either given vehicle (as used to formulate Ruxolitinib) or no treatment. Additional healthy control animals were left uninfected. The above treatments and appropriate controls were tested in three independent experiments, the results of which are summarized in Table 1. First and most importantly, Ruxolitinib-treated mice were uniformly protected against terminal disease (0/15 animals, Table 1; 0%), while 39% of untreated controls reached humane endpoints (9/23 animals, Table 1; *p* = 0.006 by Fisher’s exact test). Analogously and in keeping with our earlier report [6], none of the animals given IFN-γ blocking antibody developed terminal disease. In striking contrast, animals receiving TNF blockade exhibited an overall disease incidence rate of 40%, which was virtually identical to untreated controls (39% overall, Table 1).

In our recent study on AVHF pathogenesis in LCMV-infected HHD mice we identified hypothermia as a reliable indicator of the terminal shock syndrome [6]. In keeping with the pattern of disease incidence reported in Table 1, Ruxolitinib but not anti-TNF antibody blockade prevented hypothermia (Figure 1B). The microvascular leak as underlying pathophysiological mechanism of disease has been shown to result in albumin extravasation and the accumulation of a transudate [6]. While this earlier work suggested dysregulation of microvascular blood flow to play an important role in transudate formation [6], a potential contribution of structural alterations with consequent hyperpermeability of the endothelial barrier [20,21] remain to be investigated. Irrespective of these underlying mechanisms, which have not been the subject of the present study, the extent of fluid extravasation can be quantified by determining the skin edema (wet/dry ratio) in ventral skin flaps of mice and by assessing pleural effusions [6] to serve as surrogate of disease severity. We observed that Ruxolitinib but not TNF-blocking antibody largely normalized these disease parameters in LCMV-infected HHD mice (Figure 1C,D). Taken together, these findings indicated that Ruxolitinib prevented AVHF in our model by preventing a microvascular leak, while TNF blockade failed to afford analogous protective effects.

### 3.2. Ruxolitinib Protection in AVHF Is Unrelated to Viral Loads, Systemic NO or Hepatic iNOS Expression Levels

To mechanistically investigate the protective effects of Ruxolitinib in our model of AVHF we titrated infectious virus in blood on day eight (Figure 2A) and performed immunohistochemical stains to assess viral dissemination in the liver as an exemplary target organ of LCMV infection (Figure 2D). Neither read-out provided evidence that Ruxolitinib or anti-TNF antibody therapy interfered with viral replication or tissue dissemination. Our previous work has revealed that anti-IFN-γ therapy operates by reducing iNOS mRNA and protein expression in Kupffer cells of the liver to dampen systemic NO effects and prevent a microvascular leak [6].

To test whether the JAK1/2 inhibitor Ruxolitinib exerted analogous effects, supposedly by acting downstream of IFN-γ, we determined the totality of the NO metabolites nitrite and nitrate (“NOx”) as commonly used surrogate of serum NO (Figure 2B). NOx was clearly elevated in LCMV-infected as compared to uninfected mice, as expected [6] but surprisingly, Ruxolitinib therapy, albeit protective in our model, failed to detectably suppress serum NOx levels. Neither did anti-TNF antibody affect serum NOx levels. Hence, we tested a potential effect of Ruxolitinib on iNOS mRNA and protein expression in the liver (Figure 2C,D), the organ where we had found these parameters to be upregulated most prominently [6]. In keeping with the findings in serum (Figure 2B), iNOS mRNA was clearly elevated in LCMV-infected as compared to uninfected mice, but neither Ruxolitinib nor anti-TNF antibody treatment repressed iNOS mRNA levels (Figure 2C). Moreover, iNOS-positive cells were consistently found in periportal areas of LCMV-infected livers but not in uninfected controls, as expected (Figure 2E, [6]). However, neither Ruxolitinib nor anti-TNF exerted a clear effect on the occurrence of iNOS-expressing cells in the liver (Figure 2E). Neither treatment visibly reduced hepatic T cell infiltration (Figure 2F).

## 4. Discussion

The present results propose the repurposing of the Jak-inhibitor Ruxolitinib, an FDA-approved drug, as an experimental host-directed therapy against AVHF. Despite potential side-effects such as myelosuppression upon long-term administration [22], the drug’s safety profile and its established clinical use provide incentive for non-human primate studies and should, eventually, facilitate clinical trials and drug supply in the field.

The mechanism(s) whereby Ruxolitinib prevents microvascular leak and terminal shock in our model of AVHF deserves further investigation. We have previously reported iNOS-deficient HHD mice to be resistant against disease, positioning iNOS and systemic NO as an essential pathway and mediator of AVHF pathogenesis, respectively [6]. Moreover, we have observed that IFN-γ blockade repressed iNOS and prevented disease [6]. Accordingly, our rationale for the present evaluation of Ruxolitinib in AVHF consisted in its documented inhibition of IFN-γ-induced JAK-STAT signaling, which has been implicated in IFN-γ-mediated iNOS induction [23,24,25]. IFN-γ receptor signaling involves, however, also the MAP kinase, PI3K, CaMKII and NF-κB pathways [26], and the individual contribution of these pathways to iNOS induction remains ill-defined. Interestingly, Ruxolitinib treatment apparently prevents the microvascular leak and resulting terminal disease without detectably interfering with iNOS induction or systemic NO release. A functional impact of Ruxolitinib on antiviral T cell responses [27], which are key to the pathogenesis of LASV and LCMV in HHD mice [5,6], represents one likely mechanism, which should be investigated in future studies. Taken together with our earlier observations [6] the present findings suggest, therefore, that iNOS is necessary but not sufficient for the terminal shock syndrome in AVHF, and that additional druggable pathways are involved. On the contrary and in contrast to findings in Dengue virus infection [16,28,29], our findings do not support a key role for TNF in AVHF pathogenesis. A contribution of TNF cannot, however, be excluded.

The possible benefit of Ruxolitinib as a host-directed therapy in viral infection has precedence in COVID-19. Similar to AVHF, the most severe forms of this disease are thought to result from an excessive immune response [30,31], which manifest in a cytokine storm that can be dampened by Ruxolitinib [32]. Preliminary evidence suggests even that Ruxolitinib as well as the JAK/STAT inhibitor Baricitinib may offer clinical benefit and/or prevent progression to the most severe forms of COVID-19 [33,34,35]. Together with the present findings in a relevant small animal model these encouraging clinical signals from the ongoing COVID-19 pandemic provide renewed incentive for host-directed therapies in AVHF.

## Figures and Tables

**Figure 1 microorganisms-09-00564-f001:**
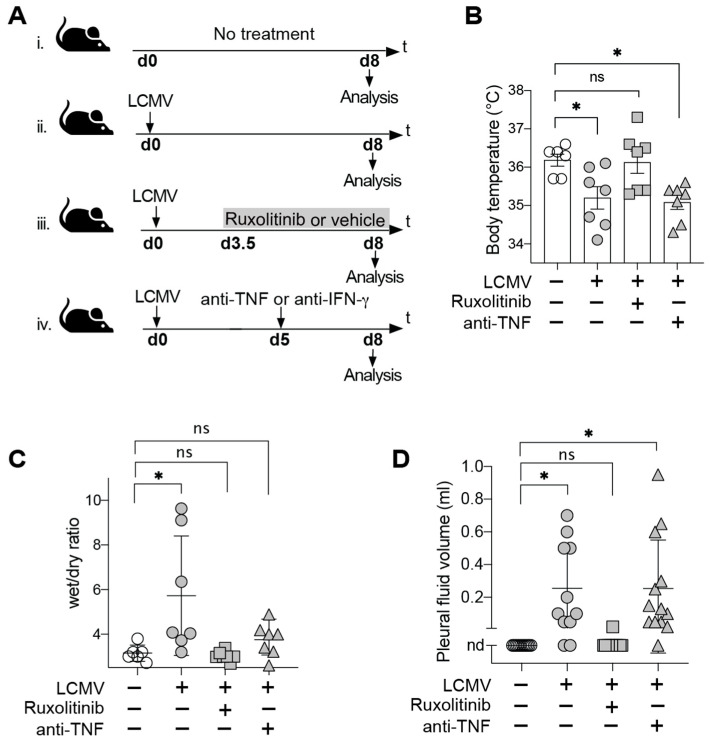
Ruxolitinib Treatment Prevents Microvascular Leak and Terminal Disease in LCMV-Infected HHD Mice. (**A**) Schematic illustration of the experimental timeline. HHD mice were infected intravenously (i.v.) with LCMV on day 0. Infected animals were treated with either Ruxolitinib (i) by oral gavage every 12 h between day 3.5 and 8 or with anti-TNF (ii) intraperitoneally on d5. Animals in the control group (iii) were left untreated. All animals were euthanized 8 days post infection in order to measure iNOS mRNA expression in liver, pleural effusions and skin edema. (**B**) Body temperature was measured on day 8. *n* = 5–8. (**C**) Wet/dry ratios of ventral skin flaps harvested on day 8 of infection. *n* = 5–8 (**D**) Pleural effusions were quantified on day 8 of infection. nd, not detectable. Symbols show individual mice. Results in (**D**) summarize data from 11–14 mice in each group from two experiments. One representative experiment (**B**,**C**) of two independent experiments is shown. * *p* < 0.05 by one-way ANOVA with Dunnett’s post-test; ns: not statistically significant.

**Figure 2 microorganisms-09-00564-f002:**
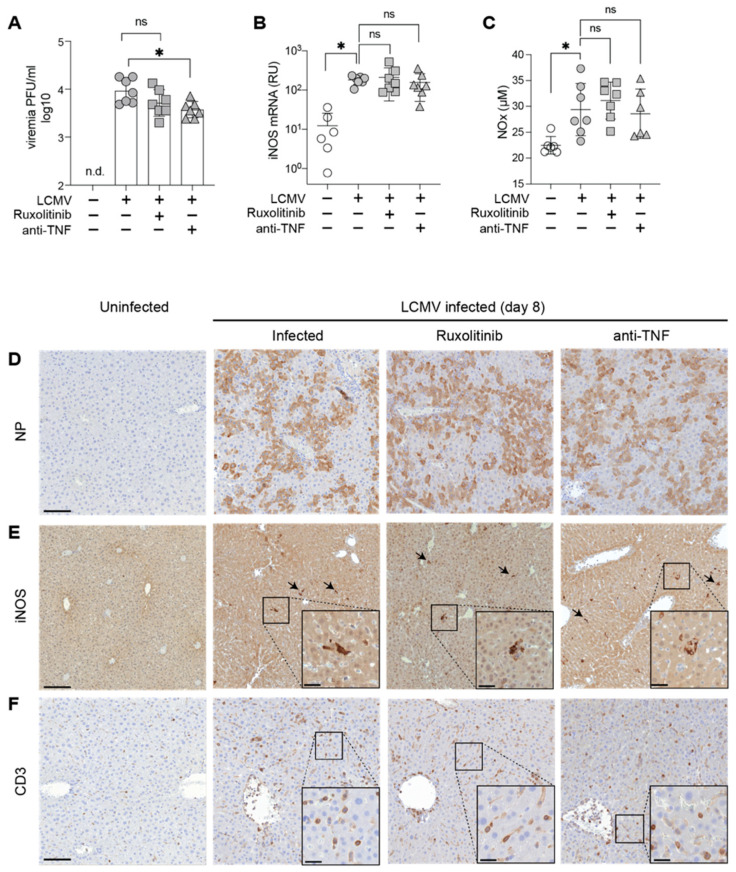
Protective Activity of Ruxolitinib is not Linked to iNOS Expression, Serum NOx Levels or Viral Load. HHD mice were infected with LCMV on day 0 and treated with Ruxolitinib or anti-TNF as in Figure 1. (**A**) iNOS mRNA (relative units, RU) in the liver. (**B**) Total combined nitrite (NO_2_^−^) and nitrate (NO_3_^−^) concentration (NOx) in serum of day-8 infected and uninfected mice. (**C**) Viremia on day 8 of infection. n.d.: not determined. (**D**–**F**) 8 days post infection, liver tissues were prepared for histological analysis and were compared to uninfected HHD mice. Sections were processed for immunohistochemical detection of LCMV NP, iNOS and T cells (CD3). Arrows indicate iNOS expressing cells. Scale bar: 100 μm. Scale bar in insets: 20 μm. Symbols representing individual animals and the mean ± SEM of 6–7 mice per group from one of two independent experiments are plotted (**A**–**C**). * *p* < 0.05 by one-way ANOVA with Dunnett’s post-test in comparison with infected; ns, not significant.

**Table 1 microorganisms-09-00564-t001:** Incidence of terminal disease in LCMV-infected HHD mice receiving various treatments.

Experiment ^a^	Treatment ^b^	Diseased Animals/Tested Animals ^c^	Day of Termination ^d^	Diseased Mice % ^e^
1	None Vehicle Ruxolitinib	5/8 6/8 0/8	8/8/10/12/12 8/8/10/10/11/11 -	62.5 75 0
2	None Ruxolitinib anti-TNF	1/8 0/8 2/8	8 - 8/8	12.5 0 25
3	None anti-TNF anti-IFN-γ	3/7 4/7 0/7	7/8/8 7/7/8/8 -	42.9 57.1 0
1–3 combined	None Ruxolitinib anti-TNF anti-IFN-γ	9/23 0/16 6/15 0/7	7/8/8/8/8/8/10/12/12 - 7/7/8/8/8/8 -	39.1 0 40 0

^a^ HHD mice were infected with LCMV in three separate experiments (1–3). They were monitored for disease onset and euthanized when reaching humane endpoints. Combined results from the same experiments (1–3) are summarized at the bottom of the table. ^b^ Mice received the indicated treatments or were left untreated (none). ^c^ The number of diseased animals (reaching humane endpoints) and the total number of animals tested are indicated. ^d^ Diseased animals were euthanized in accordance with Swiss law. Mice in experiment No. 1 were monitored for 23 days after infection, experiment No. 2 and No. 3 were terminated on day 8 after infection for additional analyses. ^e^ The percentage of diseased mice per group was calculated based on the numbers in column (c).

## Data Availability

The data presented in this study are available from the authors on request.

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
