# Peer review of "The Janus Kinase Inhibitor Ruxolitinib Prevents Terminal Shock in a Mouse Model of Arenavirus Hemorrhagic Fever"

_microorganisms, 2021, doi:10.3390/microorganisms9030564_

Round 1
Reviewer 1 Report
This is a well-written manuscript about the use of a Janus Kinase inhibitor (ruxolitinib) to treat mice for arenavirus hemorrhagic fever. This drug reduces the effects of high circulating IFN-a that tends to corrolate with bad prognosis during Argentine Hemorrhagic fever viral infection (Levis SC et al 1985). In a previous publication (Remy et al, 2017), the authors noted the importance of Nitrous Oxide in Ebola virus disease and tested the hypothesis that it might be just as lethal in Lassa fever, stating that blockade of IFN-g or depletion of T cells repressed hepatic iNOS and prevented disease despite unchecked high-level viremia. The current work somewhat dampens those claims about iNOS, since iNOS levels were low and did not correlate with disease.
While ruxolitinib can be a lifesaver for individuals suffering from GVHD, the drug's inhibition of JAK2 protein can result in myelo-suppression, primarily expressed as anemia and thrombocytopenia, and less frequently as neutropenia, which rarely leads to drug discontinuation. Its anti-JAK1 inhibitory action is responsible for the reduction of pro-inflammatory cytokines, with a consequent improvement of symptoms, quality of life and, ultimately, bone marrow fibrosis. At the same time, the anti-cytokine action could potentially cause an immunosuppressive effect, since the immune system and the hematopoietic system share intracellular signaling pathways, mediated by common receptors for cytokines and growth factors. So, this reviewer is not enthusiastic about another treatment that happens to work in mice under precise experimental conditions but that is unlikely to have positive effects in humans in the throes of viral hemorrhagic fever.
The authors chose wet/dry tissue weights for assessing vascular leakage. It would have been nice to capture an image illustrating the uptake of blue dye as a measure of barrier permeability.
On the whole, this is a well-executed series of experiments and a fair portrayal of the results.
Minor problems:
1) AHF should not be used as an abbreviation for arenavirus hemorrhagic fever because it has so commonly been used in the past as an abbreviation for Argentine hemorrhagic fever. Since the hemorrhagic fever side of this disease is relatively rare, why not just call it "arenavirus disease".
2) Fig 1 line 193, sentence ends with "day" and should end with "day 0".
3) Line 209, when you mention "previous work" please put in a reference.
4) Fig 2 legend, should change "(D)" to "(D, E, F)".
3) Ref 25 misspells gamma as gama.
Author Response
Reviewer #1
This is a well-written manuscript about the use of a Janus Kinase inhibitor (ruxolitinib) to treat mice for arenavirus hemorrhagic fever. This drug reduces the effects of high circulating IFN-a that tends to corrolate with bad prognosis during Argentine Hemorrhagic fever viral infection (Levis SC et al 1985). In a previous publication (Remy et al, 2017), the authors noted the importance of Nitrous Oxide in Ebola virus disease and tested the hypothesis that it might be just as lethal in Lassa fever, stating that blockade of IFN-g or depletion of T cells repressed hepatic iNOS and prevented disease despite unchecked high-level viremia. The current work somewhat dampens those claims about iNOS, since iNOS levels were low and did not correlate with disease.
While ruxolitinib can be a lifesaver for individuals suffering from GVHD, the drug's inhibition of JAK2 protein can result in myelo-suppression, primarily expressed as anemia and thrombocytopenia, and less frequently as neutropenia, which rarely leads to drug discontinuation. Its anti-JAK1 inhibitory action is responsible for the reduction of pro-inflammatory cytokines, with a consequent improvement of symptoms, quality of life and, ultimately, bone marrow fibrosis. At the same time, the anti-cytokine action could potentially cause an immunosuppressive effect, since the immune system and the hematopoietic system share intracellular signaling pathways, mediated by common receptors for cytokines and growth factors. So, this reviewer is not enthusiastic about another treatment that happens to work in mice under precise experimental conditions but that is unlikely to have positive effects in humans in the throes of viral hemorrhagic fever.
In response to the reviewer’s comment, we have reworked the discussion section of the manuscript to explicitly state that Ruxolitinib can have side effects such as myelosuppression, notably upon long-term administration (Line 275-276).
The authors chose wet/dry tissue weights for assessing vascular leakage. It would have been nice to capture an image illustrating the uptake of blue dye as a measure of barrier permeability.
Evans blue dye is used as a tracer for albumin to which it binds, but depending on its concentration in serum it can also penetrate tissues as a free dye in a virtually unrestricted manner (see e.g. Moos and Mollgard. Neuropathol Appl Neurobiol. 1993 Apr;19(2):120-7). In our publication by Remy et al. 2017 we have, therefore, directly quantitated the albumin content of extravascular fluid accumulations instead of using the Evans blue dye surrogate. These earlier studies have demonstrated that extravascular fluid accumulations in LCMV-infected HHD mice represent a transudate rather than an exudate. In response to the reviewer’s comment the manuscript’s text has been amended to mention this nature of the interstitial fluid accumulation (Line 204-209).
On the whole, this is a well-executed series of experiments and a fair portrayal of the results.
Minor problems:
1) AHF should not be used as an abbreviation for arenavirus hemorrhagic fever because it has so commonly been used in the past as an abbreviation for Argentine hemorrhagic fever. Since the hemorrhagic fever side of this disease is relatively rare, why not just call it "arenavirus disease".
In response to the reviewer’s critique and to avoid any potential mix-up between “Arenavirus hemorrhagic fever” and “Argentine hemorhagic fever” the revised manuscript abbreviates “Arenavirus hemorrhagic fever” as AVHF instead of AHF. We leave it to the editor’s discretion whether the term “arenavirus disease” should be used instead. We propose the abbreviation AVHF since it avoids any potential mix-up with other forms of arenavirus disease such as choriomeningitis or congenital malformation.
2) Fig 1 line 193, sentence ends with "day" and should end with "day 0".
We have inserted the missing “0” (line 223).
3) Line 209, when you mention "previous work" please put in a reference.
We have add a reference to this sentence (Remy et al. 2017) as recommended by the reviewer (Line 242).
4) Fig 2 legend, should change "(D)" to "(D, E, F)".
The figure legend has been corrected to read (“D,E,F)” (Line 250).
3) Ref 25 misspells gamma as gama.
The reference has been corrected to read “gamma” instead of “gama”.

Reviewer 2 Report
Lasa virus targets both the immune system and vascular endothelial cells (ECs), and cause a severe vascular leakage syndrome. However, the underlying mechanisms remain unclear.
In this study, Sahin, Liu et al. use mouse model infected with a BSL2 strain LCMV to repurpose Ruxolitinib in prevention of infection-induced microvascular leakage syndrome. While some of the results are interesting, there are a critical aspect missing in assessing vascular endothelial barrier function. Below, I provide specific comments to the authors that they should address in order to improve the manuscript.
Major comments
Adherens and tight junctions are intercellular junctions crucial for microvascular endothelial barrier function. The authors presented detail information of a well-established mouse model, which could support immunohistochemistry analysis of adherens (i.e. VE-cadherin, catnin, or p120) and tight (i.e. occluding, claudin, or ZO-1) as not only the structural aspect but also signaling pathway in regulation of microvascular endothelial barrier function. The authors should at least address these two junctions using tissue samples to improve the manuscript.
Minor comments
- The overall premise that differences between the BSL4 strain Lassa virus and the BSL2 LCMV strain needs to be better explained how cross species (both host and pathogen) comparisons are justified.
- Spell out LCMV when first appears in the manuscript.
- Authors should limit the use of acronyms to improve the readability and evaluation of the data. For example, LASC is not necessary in this manuscript.
Author Response
Reviewer #2
Lasa virus targets both the immune system and vascular endothelial cells (ECs), and cause a severe vascular leakage syndrome. However, the underlying mechanisms remain unclear.
In this study, Sahin, Liu et al. use mouse model infected with a BSL2 strain LCMV to repurpose Ruxolitinib in prevention of infection-induced microvascular leakage syndrome. While some of the results are interesting, there are a critical aspect missing in assessing vascular endothelial barrier function. Below, I provide specific comments to the authors that they should address in order to improve the manuscript.
Major comments
Adherens and tight junctions are intercellular junctions crucial for microvascular endothelial barrier function. The authors presented detail information of a well-established mouse model, which could support immunohistochemistry analysis of adherens (i.e. VE-cadherin, catnin, or p120) and tight (i.e. occluding, claudin, or ZO-1) as not only the structural aspect but also signaling pathway in regulation of microvascular endothelial barrier function. The authors should at least address these two junctions using tissue samples to improve the manuscript.
We acknowledge that our study has not been designed to provide new insights into the structural alterations underlying arenavirus-induced vascular leak but was aimed to test new experimental therapies. In response to the reviewer’s comment we have amended the manuscript to explicitly acknowledge that the structural changes in the endothelial barrier accounting for the microvascular leak remain to be investigated (Line 206-209).
Minor comments
- The overall premise that differences between the BSL4 strain Lassa virus and the BSL2 LCMV strain needs to be better explained how cross species (both host and pathogen) comparisons are justified.
The text has been amended to explain that LCMV-WE is genetically closely related to LASV, is not normally pathogenic for healthy human adults but causes AVHF in both non-human primates and HHD mice and therefore has been used as a model organism to investigate mechanisms of AVHF pathogenesis without the need for BSL-4 laboratory containment (Line 48-52). - Spell out LCMV when first appears in the manuscript.
We have corrected this deficiency of our manuscript by spelling out “lymphocytic choriomeningitis virus (LCMV)” the first time the virus is mentioned in our manuscript. - Authors should limit the use of acronyms to improve the readability and evaluation of the data. For example, LASC is not necessary in this manuscript.
We have revised the manuscript to minimize the use of acronyms. LASC and other rarely used abbreviations namely VHF, HLH, SPF, ATCC, FCS, HRP and PBS have been eliminated by writing them out in full.

Round 2
Reviewer 2 Report
The authors mainly claim: "Our findings position Ruxolitinib but not TNF blockade as a promising strategy for the prevention of microvascular leak in AVHF treatment......". As I explained before, information about adherens and tight junctions are crucial in addressing microvascular hyperpermeabilization. However, the authors failed in present any experimental data about this crucial issue. Since this is a research article, not a review article, a simple information inserted into discussion is far from enough to support the claim under this title. One of feasible ways is to do immunostaining using VE-cadherin, zo-1, occludin, or claudin in the mouse tissues to visualize these junctional structures.
